# Study into the reversal of septic shock with landiolol (beta blockade): STRESS-L Study protocol for a randomised trial

Ranjit Lall,[1] Dipesh Mistry,[1] Emma Skilton,[1] Nafisa Boota,[2] Scott Regan,[1] Julian Bion [orcid] ,[3] Simon Gates,[1,4] Anthony C Gordon,[5] Janet Lord,[6] Daniel Francis McAuley [orcid] ,[7] Gavin Perkins,[1,8] Mervyn Singer,[9] Duncan Young,[10] Tony Whitehouse [orcid] [11,12]

For numbered affiliations see end of article.

**Correspondence to**
Dr Tony Whitehouse;
tony.whitehouse@uhb.nhs.uk

## ABSTRACT

**Introduction** In 2013, a single-centre study reported the safe use of esmolol in patients with septic shock and tachycardia who required vasopressor therapy for more than 24 hours. Although not powered to detect a change in mortality, marked improvements were seen in survival (adjusted HR, 0.39; 95% CI, 0.26 to 0.59; p<0.001). Beta blockers are one of the most studied groups of drugs but their effect in septic shock is poorly understood; proposed mechanisms include not only the modulation of cardiac function but also immunomodulation.

**Methods and analysis** STRESS-L is a randomised, open-label, non-blinded clinical trial which is enrolling a total of 340 patients with septic shock as defined by Sepsis-3 consensus definition and a tachycardia (heart rate ≥95 beats per minute (bpm)) after vasopressor treatment of at least 24 hours. Standard randomisation (1:1 ratio) allocates patients to receive usual care (according to international standards) versus usual care and a continuous landiolol infusion to reduce the heart rate between 80 and 94 bpm. The primary endpoint is the mean Sequential Organ Failure Assessment score over 14 days from entry into the trial and while in intensive care unit. Results will inform current clinical practice guidelines.

**Ethics and dissemination** This trial has clinical trial authorisation from the UK competent authority, the Medicines and Healthcare products Regulatory Agency, and has been approved by the East of England-Essex Research Ethics Committee (reference: 17/EE/0368). The results of the trial will be reported first to trial collaborators. The main report will be drafted by the trial coordinating team, and the final version will be agreed by the Trial Steering Committee before submission for publication, on behalf of the collaboration.

**Registration** The trial is funded by the National Institute for Health Research Efficacy and Mechanism Evaluation (EME) (Project Number: EME-14/150/85) and registered ISRCTN12600919 and EudraCT: 2017-001785-14.

## INTRODUCTION

A randomised single-centre study of 154 patients from Rome[1] reported the use of beta-adrenergic blockade using the short-acting

### Strengths and limitations of this study

► STRESS-L is a randomised study comparing the use of the ultra-short acting beta blocking agent, landiolol, with standard care in patients with septic shock, persistent tachycardia treated with high-dose norepinephrine.

► STRESS-L will study 340 patients, which we believe will be the largest study of its type.

► There is little known about the effects of beta blockade and the mechanisms induced by beta-1 receptor antagonist in this population.

► Its limitation is that the study is not double blind since the dosing of the investigational medicinal product (landiolol) is variable according to a physiological endpoint (target heart rate of 80–94 beats per minute), a placebo would be ineffective and the intervention impossible to blind.

agent esmolol in patients with septic shock and tachycardia who required vasopressor therapy for more than 24 hours. The dosing endpoint was a reduction in heart rate between 80 and 94 beats per minute (bpm). Though this study was not powered to detect a change in mortality, marked improvements were seen in survival (adjusted HR, 0.39; 95% CI, 0.26 to 0.59; p<0.001). Improvements were also seen in the beta blocker-treated group in the duration of vasopressor treatment, and in measures of renal and cardiac function.

Autonomic dysfunction and tachycardia are associated with a poor outcome in septic shock[2] with mortalities in excess of 70%.[3] Bradycardia is relatively protective.[4] Currently, beta agonists are recommended as part of resuscitation and restoration of cardiac output and blood pressure in septic shock[5] despite an early study examining the

use of dobutamine to increase cardiac index and systemic oxygen delivery being stopped prematurely because of an increase in the intervention cohort.[6] Beta blockade represents a paradigm shift in management of patients with septic shock and arises from observations of harm induced by excessive sympathetic activation and benefit from beta-adrenergic blockade. Limited data from animal models[7-12] and patients[1 13 14] suggest improved cardiovascular and immune function by their use. Excess beta-adrenergic activation may be both endogenous (related to the severity and duration of the underlying infection) and exogenous due to the catecholamine therapy which has until now been the mainstay of management of hypotension caused by septic shock.[5] A combination of beta-2 agonist with beta-1 blockade improves sepsis-induced immune, cardiovascular and coagulation dysfunction in established sepsis (reviewed in[15]).

Landiolol (Rapibloc, AOP Orphan Pharmaceuticals, Vienna, Austria) is an ultra-short-acting beta blocker that has a half-life of 2.3–4 min[16] and is approximately eight times more selective for the beta-1 receptor than esmolol.[17] Landiolol is metabolised by plasma pseudocholinesterase and liver carboxylesterse to inactive metabolites and its metabolism is unaffected by liver impairment.[18] A few preclinical studies have examined landiolol in sepsis. It decreased circulating levels of the cytokines, tumour necrosis factor (TNF)-alpha, interleukin (IL)-6, and high mobility group box-1 and reduced histological lung damage in a rat endotoxin model.[11] Landiolol was also cardioprotective in rats with septic shock by normalising the expression of cardiac vasoactive peptide endothelin-1.[19] Landiolol has been used safely and successfully in a patient with septic shock with atrial fibrillation[20] and in sepsis-induced tachycardia.[21] A recent trial of 151 patients from 54 centres in Japan[22] demonstrated that landiolol could be safely administered to patients with septic shock.

Landiolol rather than esmolol was selected for this clinical trial because of its superior beta-1 specificity and the expectation that stopping the landiolol infusion would allow a rapid return to the baseline haemodynamic state if investigators deemed the intervention harmful. We will also report data of safety of landiolol infusion as part of the study.

## STUDY OBJECTIVES
### Primary objective
The primary objective of this trial is to assess the efficacy, safety and mechanisms of landiolol (beta blockade) in patients with septic shock and tachycardia requiring prolonged (>24-hour continuous treatment) support with high-dose vasopressor agents.

### Secondary objectives
The secondary objectives of the trial are:
► To determine whether infusion of the rapid-acting, ultra-short-lived and highly specific beta-adrenergic antagonist landiolol improves mortality and length of hospital stay compared with current best clinical practice, in patients who have septic shock.
► To investigate the pathways that are altered by beta blockade in septic shock by examining the effects of landiolol on blood markers of inflammation, metabolism and cardiomyocyte damage.

## METHODS AND ANALYSIS
This manuscript has been written in concordance with the Standard Protocol Items: Recommendations for Interventional Trials guidelines.[23]

### Trial design
This is a multicentre, randomised, controlled open label phase IIb trial comparing usual treatment with usual treatment plus landiolol infusion in a total of 340 patients. The trial is conducted and managed by Warwick Clinical Trials Unit and sponsored by the University Hospitals of Birmingham (UHB) National Health Service Foundation Trust. It is coordinated by a Trial Management Group and independent oversight is provided by the Trial Steering Committee and a Data Monitoring Committee.

The trial has been designed and will be reported in line with the CONSORT (Consolidated Standards of Reporting Trials) statement[24] and the conduct has been planned in full conformance with the principles of the Declaration of Helsinki and Good Clinical Practice.

### Study setting
The study recruitment commenced in May 2018. The main trial is taking place in over 35 UK adult intensive care units (ICUs). Participating ICUs will have a typical case mix for UK critical care, a track record of recruitment to clinical trials, suitable support for screening and data collection, and the means to store blood samples for transfer to the central analysis unit. They should also be willing to manage atrial fibrillation with correction of potassium, magnesium and amiodarone in the usual care group as the use of beta blockers in these patients would risk making the trial results uninterpretable.

### Inclusion criteria
Patients are eligible to be included in the trial if they meet the following criteria:
► Aged 18 years or above.
► Being treated in a critical care unit.
► Septic shock according to internationally accepted definitions.*
► Heart rate ≥95 bpm (at the time of randomisation).
► Receiving vasopressor support to maintain a target blood pressure for ≥24 hours.
► Are being treated with norepinephrine at a rate ≥0.1 µg/kg/min.
 *Sepsis-3 definitions[25]:
► Confirmed or suspected infection requiring antibiotic therapy.

- ► New organ dysfunction, as evidenced by an increase in Sequential Organ Failure Assessment (SOFA) score ≥2.
- ► A blood lactate >2 mmol/L at any point during shock resuscitation.
- ► Vasopressor therapy to maintain mean arterial pressure (MAP) ≥65 mm Hg.

In particular, the presence of a blood lactate >2 mmol/L is only necessary for the diagnosis of septic shock and is NOT necessary for randomisation 24 hours later.

### Exclusion criteria

The participant may not enter the trial if any of the following applies:

- ► Tachycardia as a result of pain, discomfort from medical devices (including endotracheal tubes), during interventions or other patient distress.
- ► Any form of vasodilatory shock that is not caused by sepsis.
- ► Norepinephrine infusion <0.1 µg/kg/min.
- ► >72 hours after start of vasopressor therapy.
- ► <12 hours since norepinephrine to treat a medical condition other than septic shock stopped.
- ► Having pre-existing severe cardiac dysfunction (New York Heart Association grade 4 or more).
- ► Having pre-existing severe pulmonary hypertension (mean pulmonary artery pressures >55 mm Hg).
- ► Acute severe bronchospasm (due to asthma or chronic obstructive pulmonary disease).
- ► Untreated second or third degree heart block.
- ► Untreated phaeochromocytoma.
- ► Prinzmetal's angina.
- ► A history of ischaemic stroke or transient ischaemic attack or untreated severe carotid stenosis.
- ► Advanced liver disease with Child-Pugh score of ≥B.
- ► Known sensitivity to beta blockers.
- ► Patient/legal representative unwilling to provide written informed consent.
- ► Known to be pregnant.
- ► Terminal illness other than septic shock with a life expectancy <28 days.
- ► Participants who have been administered an investigational medicinal product (IMP) for another research trial in the past 30 days.
- ► Patients in whom the clinical team feel are about to finish their norepinephrine therapy.
- ► Decision of withdrawal of care is in place or imminently anticipated.
- ► Receiving extracorporeal membrane oxygenation treatment.

Co-enrolment of study participants onto other interventional studies will be considered where there is no possible conflict with the STRESS-L trial objectives.

### Consent

Eligible patients who are deemed competent will be provided with a copy of the patient information sheet before informed consent is sought. However, due to the nature of the underlying condition and its treatment, many patients will be unable to give informed consent. The vulnerability of this patient group is fully appreciated and every effort must be undertaken to protect their safety and well-being. To ensure this, consenting will be obtained in accordance with the Medicines for Human Use Regulations and the Health Research Authority ethics guide on medical research involving adults who cannot consent (http://www.hra.nhs.uk/resources/before-you-apply/consent-and-participation/adults-unable-to-consent-for-themselves/).

### Intervention

Participants will be randomised to either usual care (control group) or usual care and landiolol (intervention group).

### Usual care

Participants randomised to this arm will receive usual care for septic shock, namely:

- ► Timely treatment of the source of sepsis (eg, drainage of infected fluid collections).
- ► Prompt and appropriate empirical antibiotic treatment, and modification, if needed, on the basis of culture results.
- ► Appropriate fluid resuscitation to correct hypovolaemia.
- ► The use of vasopressors to achieve a target MAP (suggested target 65–70 mm Hg).

Adequate fluid resuscitation will be achieved using repeated fluid challenges to a guided target; the use of a cardiac output monitor is not mandated. The usual care group will not receive any beta blockade for the duration of their ICU stay—if the treating clinician deems beta blockade necessary, this will be captured on the Case Report Form (CRF) and reported as a protocol deviation. All other general ICU management will be based on the latest guidance from the surviving sepsis campaign[5] and the UK national critical care guidelines (eg, ventilator and central line care bundles).

### Usual care and landiolol

An intravenous infusion of landiolol will be started at 1.0 µg/kg/min and will progressively increase every 15 min at increments of 1.0 µg/kg/min, to reach the target heart rate of 80–94 bpm usually within 6 hours. Landiolol will be administered peripherally or centrally but must be on a dedicated line. Landiolol has an elimination half-life of 2.3–4 min[17] and so a loading dose is unnecessary. The landiolol infusion will be continued until the pulse rate is persistently below 95 bpm.

While a patient is receiving vasopressor agents (norepinephrine, vasopressin), the landiolol infusion will be adjusted accordingly to maintain the target heart rate of 80–94 bpm as per the landiolol infusion protocol (figure 1: STRESS-L Study drug infusion protocol). Once the patient is consistently within the target heart rate of 80–94 bpm, the landiolol infusion will be continued and

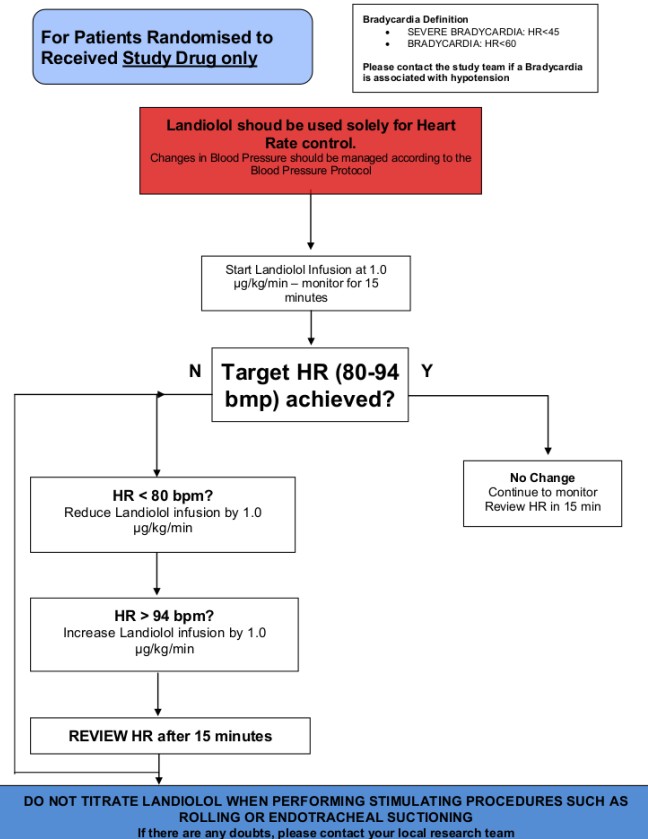

**Figure 1** STRESS-L Study drug infusion protocol. bpm, beats per minute; HR, heart rate.

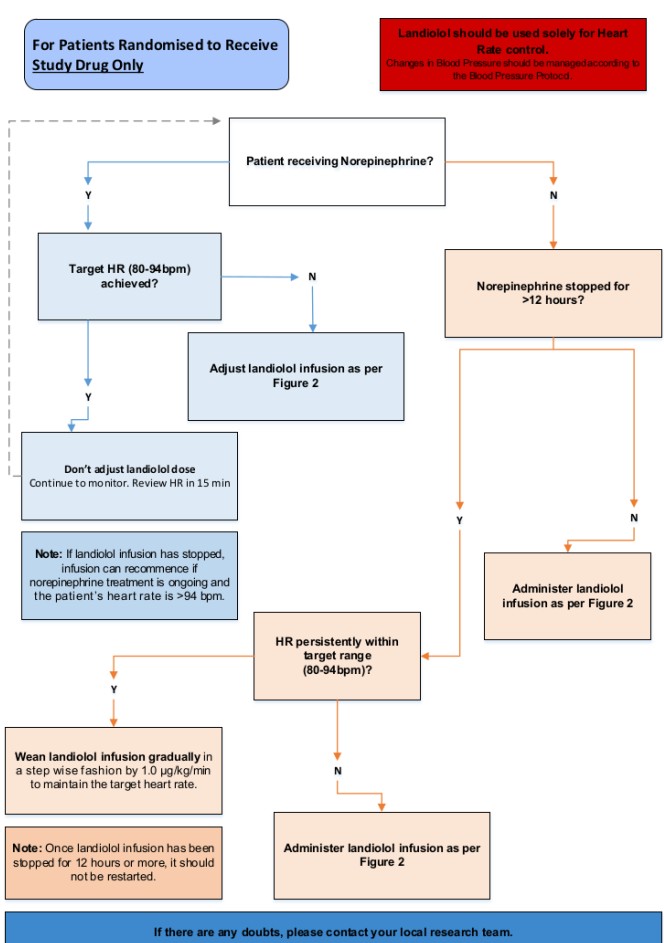

**Figure 2** STRESS-L protocol for weaning and stopping of study drug. bpm, beats per minute; HR, heart rate.

not be adjusted. The landiolol should be weaned and, if necessary, stopped while the heart rate is below 80 bpm. Once all vasopressor agents have been discontinued:

► If vasopressors have been discontinued for less than 12 hours, the landiolol infusion will continue according to figure 1.

► If vasopressors have been discontinued for more than 12 hours, the landiolol should be actively weaned (this is the end of norepinephrine treatment (EONT) timepoint) and according to figure 2: Timing and weaning of the study drug.

This trial allows for up to 14 days of landiolol treatment per participant.

## OUTCOME MEASURES
### Primary outcome
The primary outcome will be the mean SOFA score over the first 14 days from entry into the trial and while in ICU. Measurement of the SOFA score will cease if the patient withdraws from the trial, dies or is discharged from the ICU. It is assumed that delayed discharges will be evenly distributed between each group.

### Secondary outcomes
► Mortality at day 28 and day 90.
► Length of ICU and hospital stay.

► Reduction in dose and duration of vasopressor treatment (total daily administered doses).

### Mechanistic outcomes
Serial blood samples will be collected from patients and assays will include markers of myocardial dysfunction and inflammation.

► Measurement of total catecholamine. It is unknown whether beta blockade acts through altering the effects of extraneous catecholamines administered as treatment of septic shock or by modulating those produced by the patient. Serum catecholamines will be analysed in the context of the dose of inotropes being administered.

► Markers of myocardial dysfunction. Serum B-type natriuretic peptide has been demonstrated to be a reliable biomarker of myocardial injury, ischaemia and dysfunction in patients with septic shock and also as a prognostic marker for a poor outcome.[26 27] Serial measurements of troponin-T will be made.

► Measurement of serum free fatty acids and markers of fatty acid metabolism.

► Biomarkers of systemic inflammation. This will be measured using a multiplex inflammatory biomarker assay depending on the available technology at

the time of analysis. A selection of cytokines will be analysed, all or some of which will include IL-1 beta, IL-2, IL-4, IL-5, IL-6, IL-8, IL-10, IL-12, TNF-alpha, TNF-beta and interferon-gamma. This focused array will allow an assessment of the pro/anti-inflammatory balance over time in patients with septic shock and allow more detailed study of the other potential mechanisms of action of landiolol. Cortisol assays will measure the influence of beta blockade on the adrenal cortex.

► In addition, samples will be stored for subsequent analysis (eg, genetics/proteomics/metabolomics) in order to investigate early cellular responses during the resolution of sepsis.

### Safety outcomes

► The episodes of bradycardia (heart rate <50 bpm), bradycardia with haemodynamic compromise requiring intervention, significant hypotension requiring intervention (not including temporarily stopping the infusion), heart block, arrhythmia and arrhythmia with haemodynamic compromise requiring intervention will be reported.

### Sample size

The primary outcome is the mean SOFA score over the first 14 days in ICU. Using preliminary collected data from 324 patients from UHB satisfying the trial eligibility criteria, the mean SOFA score over the first 14 days in ICU was 6.3, with SD 2.4. Assuming (conservatively) an SD of 2.8, and a difference of 1 point between the beta blocker and usual care groups, obtaining a p value less than 0.05 (two-sided) with 90% power would require outcome data on 330 patients. To allow for 3% withdrawals and losses, the proposed sample size is 340 (figure 3: STRESS-L CONSORT flow diagram).

### Assignment of interventions

#### Randomisation

We will use a computerised minimisation randomisation system, created by Warwick Clinical Trials Unit in accordance with their standard operating procedure and held on a secure server, with the allocation generated per individual (participant). Participants will be randomised strictly sequentially as they become eligible for randomisation using a 1:1 ratio. The randomisation will be stratified by recruiting site and norepinephrine dose where the dose reflects the participant's severity. The dose will be dichotomised using a value of 0.3 µg/kg/min (ie, ≥0.1–0.3 µg/kg/min and >0.3 µg/kg/min) taken from the LeoPARDS trial as an indicator of low and high severity to ensure participant severity is balanced across both arms.[28]

Once written informed consent has been obtained, a member of the local research team will use an interactive voice response application to randomise. Eligibility will be confirmed by an investigator prior to randomisation. Sites will only be given access to the application once they

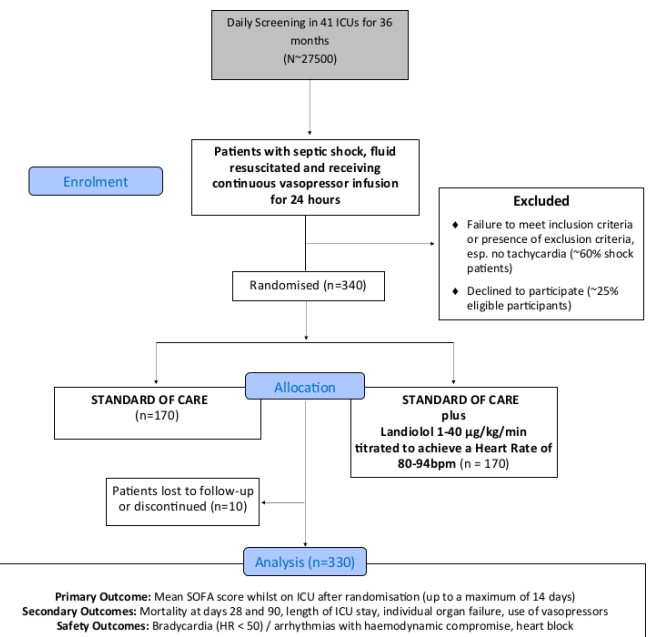

**Figure 3** STRESS-L CONSORT flow diagram. bpm, beats per minute; CONSORT, Consolidated Standards of Reporting Trials; HR, heart rate; ICU, intensive care unit; SOFA, Sequential Organ Failure Assessment.

have been given the 'green light' to begin recruitment and all required approvals are in place.

### Blinding

This an open label trial with no blinding of the treatment allocation. Blinding is not possible due to the requirement for the landiolol dose to be titrated to achieve a target heart rate.

## DATA COLLECTION, MANAGEMENT AND ANALYSIS
### Data collection
#### Screening

All patients with septic shock started on norepinephrine in ICU will be screened for trial eligibility. Once eligibility criteria are met there is a 48-hour window for randomisation. Due to this short window, informed consent may be sought during the first 24 hours of norepinephrine therapy. This will allow more time if legal representative consent is required. Randomisation will not occur until vasopressor therapy has been running for ≥24 hours, the patient is been treated with norepinephrine at rate >0.1 µg/kg/min and has a tachycardia with a heart rate ≥95. It will be made clear to the patient or their legal representative that if they no longer meet the eligibility criteria after the 48-hour mark (ie, 72 hours after the start of vasopressor therapy), the participant will not be randomised.

#### Baseline visit 24 hours prior and up to randomisation

Table 1 summarises the schedule of events for the trial. The patient's medical history, physical examination, basic demographic data and steroid use will be recorded. The participant will be randomised, and if allocated to the

**Table 1** STRESS-L schedule of interventions

| Procedure (time (T) in days/hours) | Screening (T0–12) | Baseline (day 0–24 hours – T0) | Day 1 (T0+24) | Day 2 | Day 3 | Day 4 | Day 5 | Day 6 | Day 7 | Day 8 | Day 9 | Day 10 | Day 11 | Day 12 | Day 13 | Day 14 | EONT visit | FU visit day 28 | Final visit day 90 |
|---|---|---|---|---|---|---|---|---|---|---|---|---|---|---|---|---|---|---|---|
| Eligibility assessment | ● | | | | | | | | | | | | | | | | | | |
| Informed consent | | ● | | | | | | | | | | | | | | | | | |
| Randomisation | | ● | | | | | | | | | | | | | | | | | |
| Demographics | | ● | | | | | | | | | | | | | | | | | |
| Medical history | | ● | | | | | | | | | | | | | | | | | |
| ECG | | ● | According to clinical need or if adverse event (AE)/serious AE (SAE) | | | | | | | | | | | | | | | | |
| Pregnancy test | | ● | | | | | | | | | | | | | | | | | |
| IMP | | Dispense | | | | | | | | | | | | | | End | | | |
| Blood sample | | ● | ● | ● | | ● | | ● | | | | | | | | | ● | | |
| Biobank blood sample (optional) | | ● | ● | | | | | | | | | | | | | | ● | | |
| Transport of stored serum | | | | | | | | | | | | | | | | | Batch | | |
| Local laboratory tests (normal clinical care): | | ● | ● | ● | ● | ● | ● | ● | ● | ● | ● | ● | ● | ● | ● | ● | ● | | |
| C reactive protein | | ● | ● | ● | ● | ● | ● | ● | ● | ● | ● | ● | ● | ● | ● | ● | ● | | |
| Glucose | | ● | ● | ● | ● | ● | ● | ● | ● | ● | ● | ● | ● | ● | ● | ● | ● | | |
| Lactate | | ● | ● | ● | ● | ● | ● | ● | ● | ● | ● | ● | ● | ● | ● | ● | ● | | |
| Worst PaO$_2$/FiO$_2$ | | ● | ● | ● | ● | ● | ● | ● | ● | ● | ● | ● | ● | ● | ● | ● | ● | | |
| Platelets | | ● | ● | ● | ● | ● | ● | ● | ● | ● | ● | ● | ● | ● | ● | ● | ● | | |
| Creatinine | | ● | ● | ● | ● | ● | ● | ● | ● | ● | ● | ● | ● | ● | ● | ● | ● | | |
| Bilirubin | | ● | ● | ● | ● | ● | ● | ● | ● | ● | ● | ● | ● | ● | ● | ● | ● | | |
| White cell count | | ● | ● | ● | ● | ● | ● | ● | ● | ● | ● | ● | ● | ● | ● | ● | ● | | |
| Liver function tests (ALT or AST) | | ● | ● | ● | ● | ● | | ● | | | | | | | | | ● | | |
| Central venous blood gas (BG)/arterial BG | | ● | ● | ● | ● | ● | | ● | | | | | | | | | ● | | |
| Microbiology results from local lab | | ● | ● | ● | ● | ● | ● | ● | ● | ● | ● | ● | ● | ● | ● | ● | ● | | |
| Heart rate (hourly: T0+7 days) | | ● | ● | ● | ● | ● | ● | ● | ● | | | | | | | | | | |
| Atrial fibrillation (hourly: T0+7 days) | | ● | ● | ● | ● | ● | ● | ● | ● | | | | | | | | | | |

Continued

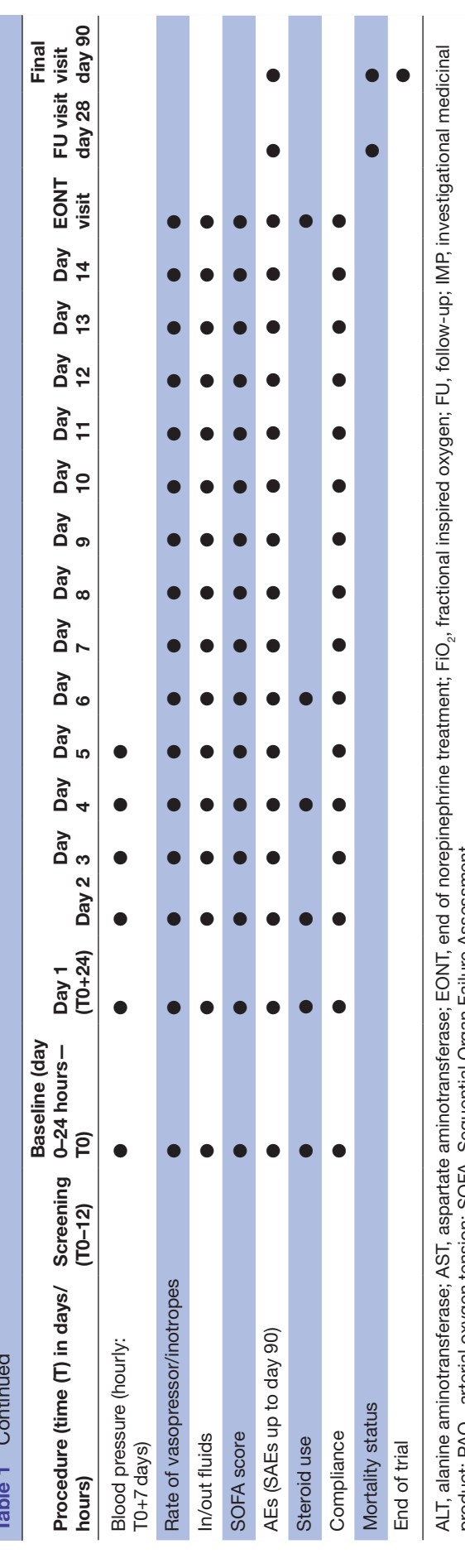

**Table 1** Continued

| Procedure (time (T) in days/hours) | Screening (T0−12) | Baseline (day 0–24 hours− T0) | Day 1 (T0+24) | Day 2 | Day 3 | Day 4 | Day 5 | Day 6 | Day 7 | Day 8 | Day 9 | Day 10 | Day 11 | Day 12 | Day 13 | Day 14 | EONT visit | FU visit day 28 | Final visit day 90 |
|---|---|---|---|---|---|---|---|---|---|---|---|---|---|---|---|---|---|---|---|
| Blood pressure (hourly): T0+7 days) | | ● | ● | ● | ● | ● | ● | | | | | | | | | | | | |
| Rate of vasopressor/inotropes | | ● | ● | ● | ● | ● | ● | ● | ● | ● | ● | ● | ● | ● | ● | ● | ● | | |
| In/out fluids | | ● | ● | ● | ● | ● | ● | ● | ● | ● | ● | ● | ● | ● | ● | ● | ● | | |
| SOFA score | | ● | ● | ● | ● | ● | ● | ● | ● | ● | ● | ● | ● | ● | ● | ● | ● | | |
| AEs (SAEs up to day 90) | | ● | ● | ● | ● | ● | ● | ● | ● | ● | ● | ● | ● | ● | ● | ● | ● | ● | ● |
| Steroid use | | ● | ● | ● | ● | ● | | ● | | | | | | | | ● | ● | | |
| Compliance | | ● | ● | ● | ● | ● | ● | ● | ● | ● | ● | ● | ● | ● | ● | ● | ● | | |
| Mortality status | | | | | | | | | | | | | | | | | ● | ● | ● |
| End of trial | | | | | | | | | | | | | | | | | | | ● |

ALT, alanine aminotransferase; AST, aspartate aminotransferase; EONT, end of norepinephrine treatment; FiO$_2$, fractional inspired oxygen; FU, follow-up; IMP, investigational medicinal product; PAO$_2$, arterial oxygen tension; SOFA, Sequential Organ Failure Assessment.

landiolol arm, IMP will be dispensed and the infusion started. As there is no checking and preparation of drug in the usual care arm, the landiolol infusion should be started within 1 hour of randomisation. The elements of the SOFA score will be recorded along with routine clinical data (cardiovascular, respiratory and renal physiological variables as well as haematological, biochemical and microbiological sample test results). The rates of norepinephrine infusion will be recorded hourly. The site and types of pathogens isolated following admission should be recorded on the electronic CRF (eCRF).

### Day 1 (time of randomisation to post-24 hours) up to day 14

The elements of the SOFA score will be recorded in the eCRF daily while the patient remains in the ICU from randomisation. This may be performed retrospectively but participating centres should ensure that blood has been sent to the local hospital laboratories for the domains that depend on laboratory tests (liver, renal and coagulation). If possible, blood should be taken for C-reactive protein. The rates of norepinephrine infusion and landiolol infusion (if randomised to this group) will be recorded hourly until day 2 to allow comparison of norepinephrine dosing and then 6 hourly thereafter. Heart rate data will be collected to allow assessment of landiolol infusion compliance and separation between groups.

Other clinical data collected will be haemodynamics, presence of atrial fibrillation, respiratory variables (blood gases with the worst P/F ratio, type of ventilator support and previous 24 hours' intravenous fluid intake, urine output and need for renal replacement therapy (as given in table 1).

### Follow-up visits (days 28 and 90)

Participants will also be followed up to ascertain survival status at 28 days and at 90 days post-randomisation. The participant's community physician will be contacted in the first instance to ascertain if the patient is alive; the participant may then be contacted by telephone. Serious adverse events (SAEs) will be reported up to day 90 following randomisation.

### Blood samples

Research blood samples will be collected on days 0, 1, 2, 4 and 6 and the EONT visit (if this does not fall on a blood sampling day). Day 0 blood samples must be taken prior to the start of landiolol infusion. Days 1, 2, 4, 6 and EONT blood samples can be taken when is convenient within the 24-hour time period. The plasma will be removed and stored as per detailed instructions provided in the trial laboratory manual. These research blood samples are mandatory as the results are required to answer the secondary outcomes of the trial.

### Statistical analysis

The primary analysis will be conducted according to intention to treat, comparing all those allocated to beta blocker (landiolol) with all of those allocated to

usual care, regardless of treatment received. Regression models will be used to estimate the treatment effects (with 95% CIs), and the models will be adjusted for clinically important covariates. Subgroup effects for baseline severity measured using the dose of norepinephrine (norepinephrine ≥0.1–0.3 µg/kg/min vs >0.3 µg/kg/min) and use of beta blockers at baseline (Yes/No) will be assessed using formal statistical tests for interaction for the primary outcome and mortality. If there is substantial non-compliance, we will conduct (complier average causal effect) analyses to estimate the treatment effect among those who received the treatment as allocated. The DMC will meet every 6 months to closely monitor the accumulating data, focusing on safety. A detailed statistical analysis plan (SAP) will be written by the study statistical team, and then finalised and approved by the DMC before any analysis is undertaken.

### Missing data
Every effort will be made to minimise missing outcome data in this trial. Further exploratory analyses to assess the impact of any missing outcome data on the SOFA score will be examined using multiple imputation techniques and other methods specified in the SAP.

### Data monitoring
The Warwick Clinical Trials Unit will be responsible for trial monitoring and visits will be conducted in accordance with the monitoring plan. On-site monitoring visits during the trial will check the accuracy of data entered into the clinical trial database against the source documents, adherence to the protocol, procedures and Good Clinical Practice, and the progress of patient recruitment and follow-up.

### SAFETY, ETHICS AND DISSEMINATION
### Adverse outcomes
All SAEs and suspected unexpected SAEs occurring from the time of randomisation to the final follow-up visit at day 90 will be recorded on the CRF and faxed or emailed to the coordinating centre within 24 hours of the research staff becoming aware of the event. In particular, bradycardia (heart rate <50 bpm) with haemodynamic compromise requiring intervention, heart block, significant hypotension requiring intervention and arrhythmia with haemodynamic compromise requiring intervention should be recorded. AE/reactions will be assessed for seriousness and reported in accordance with Medicines and Healthcare products Regulatory Agency (MHRA) guidelines.

### Regulatory and ethics approvals
The East of England-Essex Research Ethics Committee (REC: 19/EE/0368—flagged for trials involving clinical trials in patients without capacity) and MHRA have approved the study protocol. The study will comply with the principles for sharing clinical trial data from publicly funded clinical trials.

### Dissemination
The study will be reported in accordance with the CONSORT guidelines.[24] The study findings will be presented at national and international meetings with abstracts online. Presentation at these meetings will ensure that results and any implications quickly reach all of the UK and international intensive care communities. In accordance with the open access policies proposed by the National Institute for Health Research, we aim to publish the clinical findings of the trial in high-quality peer-reviewed open access (via PubMed) journals. Finally, an ongoing update of the trial will also be provided on the Warwick Clinical Trials Unit website and social media platforms, for example, Twitter (@STRESSL_trial).

### Confidentiality
In order to maintain confidentiality, all CRFs, questionnaires, study reports and communication regarding the study will identify the patients by the assigned unique trial identifier and initials only. Patient confidentiality will be maintained at every stage and will not be made publicly available to the extent permitted by the applicable laws and regulations.

### Patient and public involvement
This application is informed through a series of meetings with survivors and carers of patients who have had septic shock. Patient and public contributed to the trial proposal, protocol development and design of patient-facing materials. During the conduct of the trial, a patient and public involvement (PPI) member has been part of the Trial Management Group and has reviewed patient-facing documentation prior to the ethics and regulatory submissions and their comments have been incorporated. Two PPI representatives will sit on the Trial Steering Committee and will provide input from a patient perspective at trial meetings. Both representatives will review and provide feedback on all relevant project documents.

### DISCUSSION
Many large multicentre trials of sepsis and septic shock have not derived statistical differences between study groups following single-centre pilot studies suggesting large mortality improvements.[29–31] This is possibly because of the wide variation in severity of illness included in those studies. STRESS-L includes patients who are predicted to be at risk of a high mortality because they remain tachycardic (heart rate ≥95 bpm) after 24 hours of norepinephrine treatment. The persistence of tachycardia may phenotypically define a genetically different cohort as suggested in the population with acute respiratory distress syndrome.[32] The trial uses landiolol because of its exceptional short half-life and highly specific beta-1 receptor blockade.

The primary outcome measure (mean SOFA score), which correlates with survival, allows measurement of the effect of landiolol on organ function compared with usual care. Analysis of each domain in the SOFA score will also begin to allow us to distinguish whether alteration in cardiac efficiency is the sole mechanism for differences between groups. The collection and analysis of blood samples will also characterise further the role of the adrenergic system in septic shock and propose genetic predisposition to this specific group of patients with septic shock.

STRESS-L recruited its first patient in April 2018 but a series of protocol amendments (owing to a complex intervention) and the COVID-19 pandemic have delayed the submission of this protocol paper. As of submission, there have been 109 patients recruited, approximately 32% of its target; no patients with COVID-19 have been recruited. Our SAP will address how patients with COVID-19 (if any) will be analysed. Unpublished data suggest that the inclusion criteria for STRESS-L are rarely achieved in primary COVID-19 illness; the presence of shock (defined by consensus[25]) coincidental with high-dose norepinephrine and a tachycardia, all within the time constraints, is seen in fewer than 1% of patients. Furthermore, our sites have had most of their research staff withdrawn to clinical duties so that screening and recruitment for the study has been suspended during period of exceptionally high admissions. We expect that the large majority of patients recruited to STRESS-L will be treated for non-COVID septic shock.

**Author affiliations**
[1]Warwick Clinical Trials Unit, University of Warwick, Coventry, UK
[2]Leicester Clinical Trials Unit, Leicester, UK
[3]Intensive Care Medicine, University of Birmingham, Birmingham, UK
[4]Cancer Research UK Clinical Trials Unit, University of Birmingham, Birmingham, UK
[5]Division of Anaesthetics, Pain Medicine and Intensive Care, Imperial College London, London, UK
[6]NIHR Surgical Reconstruction and Microbiology Research Centre, University of Birmingham, Birmingham, UK
[7]Centre for Experimental Medicine, Queen's University Belfast, Belfast, UK
[8]University Hospitals Birmingham NHS Foundation Trust, Birmingham, UK
[9]Bloomsbury Institute of Intensive Care Medicine, University College London, London, UK
[10]Kadoorie Centre for Critical Care Research, Nuffield Division of Anaesthesia, University of Oxford, Oxford, UK
[11]Department of Critical Care and Anaesthesia, University Hospitals Birmingham NHS Foundation Trust, Birmingham, UK
[12]Institute of Inflammation and Aging, University of Birmingham, Birmingham, UK

**Acknowledgements** The authors would like to thank our patient and public involvement (PPI) group and AOP Orphan, Vienna, Austria, for providing the landiolol for the trial free of charge. ACG is the deputy director, and GP and DFM are co-directors of Research for the Intensive Care Foundation. We are grateful for the infrastructure provided by the National Institute of Health Research (NIHR) Critical Care Specialty Group of the Comprehensive Clinical Research Network. We would like to express our gratitude to the members of the Trial Steering Committee and the Data Monitoring Committee who provide support and guidance for this study.

**Contributors** TW and JB conceived the initial trial concept and helped develop the trial design and protocol. RL, DM and SG carried out the power calculations; and with ACG, JL, GP, DFM, MS and DY helped develop the trial design and protocol.

NB, ES and SR oversaw the governance of the protocol. TW is the chief investigator and takes overall responsibility for all aspects of trial design, the protocol and trial conduct. All authors read and approved the final manuscript.

**Funding** This project was funded by the Efficacy and Mechanism Evaluation (EME) Programme and managed by the NIHR (Project Number: 14/150/85). ACG is funded by an NIHR Clinician Scientist Fellowship. MS and GP are supported by NIHR senior investigators.

**Disclaimer** The views expressed in this publication are those of the author(s) and not necessarily those of the EME, NHS, NIHR or the Department of Health.

**Competing interests** TW reports grants from NIHR Efficacy and Mechanism Evaluation for the funding of STRESS-L (Project Number: EME-14/150/85) and during the conduct of the study; personal fees and non-financial support from AOP Orphan, manufacturer of Landiolol, outside the submitted work. MS has received travel expenses from AOP Orphan for delivering lectures. MS reports grants and other from NewB, grants from DSTL, other from Amormed, other from Biotest, other from GE, other from Baxter, grants from Critical Pressure, grants from Apollo Therapeutics, other from Roche, other from Bayer, other from Shionogi, outside the submitted work. ACG reports receiving grants from the NIHR and the NIHR Research Professorship; non-financial support from the NIHR Clinical Research Network and the NIHR Imperial Biomedical Research Centre during the conduct of the study; and personal fees from GlaxoSmithKline and Bristol-Myers Squibb outside the submitted work. DFM reports a grant from the NIHR EME programme for this study. Outside the submitted work, DFM reports personal fees from consultancy for GlaxoSmithKline, Boehringer Ingelheim, Bayer, Novartis and Eli Lilly. In addition, his institution has received funds from grants from the NIHR, Wellcome Trust, Innovate-UK and others. In addition, DFM has a patent issued to his institution for a treatment for ARDS. DFM is a director of the Research for the Intensive Care Society and NIHR EME programme director. GP reports grants from National Institute for Health Research, during the conduct of the study; and he is a director of the Research for the Intensive Care Society. All other authors declare they have no competing interests.

**Patient and public involvement** Patients and/or the public were involved in the design, or conduct, or reporting, or dissemination plans of this research. Refer to the Methods section for further details.

**Patient consent for publication** Not required.

**Provenance and peer review** Not commissioned; externally peer reviewed.

**ORCID iDs**
Julian Bion http://orcid.org/0000-0003-0344-5403
Daniel Francis McAuley http://orcid.org/0000-0002-3283-1947
Tony Whitehouse http://orcid.org/0000-0002-4387-3421

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
