## [Reviewer comments · BMJ Open]

ARTICLE DETAILS

TITLE (PROVISIONAL)	Study into the Reversal of Septic Shock with Landiolol (Beta Blockade): STRESS-L. Study Protocol for a Randomised Trial
AUTHORS	Lall, Ranjit; Mistry, Dipesh; Skilton, Emma; Boota, Nafisa; Regan, Scott; Bion, Julian; Gates, Simon; Gordon, Anthony; Lord, Janet; McAuley, Daniel; Perkins, Gavin; Singer, Mervyn; Young, Duncan; Whitehouse, Tony

VERSION 1 – REVIEW

REVIEWER	Peter Pickkers Radboudumc, The Netherlands
REVIEW RETURNED	05-Oct-2020

GENERAL COMMENTS	With interest I read the manuscript "Study into the Reversal of Septic Shock with landiolol (Beta Blockade): Study Protocol for the STRESS-L Randomised Trial". It is a study protocol for the investigation of the use of landiolol in septic shock patients. As the drawbacks of catecholamine use are becoming more apparent and better characterized, the use of beta blockers in sepsis patients is an important research topic. This study is timely and well described. I do have some comments. Background - Page 7, lines 30-3: please add the reference to the recent work by Stolk et al. who report improved immune competence in septic shock patients on beta-blockers.- Page 7, lines 36-40: This statement appears to reflect actual data, but the authors refer to a review.- Page 8, lines 6-7: "baseline cardiovascular" should be changed to "baseline hemodynamic situation" Methods - Why are patients included in which noradrenaline has been stopped (<12 hours), this seems to not correspond with the inclusion criteria (i.e. requiring vasopressors)?- The study will include patients admitted well into 2020. Will patients suffering from COVID-19 who fulfill the inclusion criteria also be included? COVID-19 has been demonstrated to have specific inflammatory properties unlike other infections/sepsis causes, which might confound results. Specific statements on this should be made in the manuscript.- Will vasopressors other than noradrenaline be used in the patients included during this trial? Outcome measures - Use of beta-blockers was associated with an enhanced TNF/IL-10 ratio in septic shock patients in the aforementioned recent
---

	study by Stolk et al. Therefore, I would advise to add this specific ratio to the outcome measures. Furthermore, noradrenaline inhibited production of CXCL-10 and MCP-1 in humans in vivo in this study. Therefore, I would advise to add these chemokines to the list of inflammatory markers of interest.  - It would be nice to add secondary ICU-acquired infections as an outcome measure. - If possible, measurements of monocytic HLA-DR expression could be added (in a subset of patients). Statistical analysis  - Heart rate is not part of the SOFA score and SOFA therefore represents a nicely chosen primary outcome - Will the detailed statistical plan be published before study inclusion is complete? Could the SAP be published as a supplement?
--	--

REVIEWER	Lex van Loon Australian National University, Canberra, Australia
REVIEW RETURNED	09-Oct-2020

GENERAL COMMENTS	Highly relevant study. Best of luck with the study. Looking forward to the results.
---

REVIEWER	FILIPPO SANFILIPPO Azienda Ospedaliero Universitaria Policlinico San Marco, Catania, Italy
REVIEW RETURNED	11-Oct-2020

GENERAL COMMENTS	Dear Editor, Thanks for the opportunity to review the protocol of the "Study into the Reversal of Septic Shock with landiolol (Beta Blockade): Study Protocol for the STRESS-L Randomised Trial". The study is promoted by a knowledgeable panel of authors and I applaud their efforts. Here my suggestions:  - I encourage the authors mention the results of the trial recently published in Lancet Resp Medicine 2020 https://pubmed.ncbi.nlm.nih.gov/32243865/ - Reference 13 (review published in 2009) may be substituted by a more recent and systematic review on beta-blockade in sepsis (i.e. https://pubmed.ncbi.nlm.nih.gov/26121122/) - Primary Objective. Please be more specific when mentioning "the efficacy". One can think of many.... Reducing SOFA, HR, other? in which timeframe? I see you later explained it, but it is better to be specific from the beginning - Secondary objective. Please change "improves" with "reduces". It is more appropriate for mortality and LOS - Regarding the "second bullet" of secondary objectives. It is better to specify if you have an hypothesis on that, explaining what exactly you suppose to find on metabolism, inflammation and cardiac samage - Study setting. I would explain readers why you are publishing a study protocol after 2 and half years. - Outcome measures. Please explain how will you handle (statistically) the SOFA score for deceased or discharged patients - Please explain why you have not considered echocardiographic evaluation as part of the study objectives. The evaluation of cardiac performance is of utmost importance when considering
--

	beta-blockade, though the identification of beta-blocker responders (non-compensatory tachycardia) and non-responders (compensatory for 'fixed' stroke volume [SV]) is challenging. Echocardiography may be particularly helpful in understanding the effects of beta-blockers in septic shock patients. Even if not performing echocardiography, other measures may be very interesting. Recently Prof Morelli in a post-hoc analysis tested the ability of the difference between systolic and diastolic pressure (SDPdifference), which reflects the coupling between myocardial contractility and a given afterload, in discriminating the origin of tachycardia. The authors found that a decrease in SDPdifference could discriminate between compensatory and non-compensatory tachycardia, revealing a covert loss of myocardial contractility not detected by conventional echocardiographic parameters and deteriorating after HR reduction with esmolol. Link: https://pubmed.ncbi.nlm.nih.gov/32690246/
--	--

VERSION 1 – AUTHOR RESPONSE

Reviewer: 1

Reviewer Name

Peter Pickkers

Institution and Country

Radboudumc, The Netherlands

Competing interests

None declared

Comments to the Author

With interest I read the manuscript "Study into the Reversal of Septic Shock with landiolol (Beta Blockade): Study Protocol for the STRESS-L Randomised Trial". It is a study protocol for the investigation of the use of landiolol in septic shock patients. As the drawbacks of catecholamine use are becoming more apparent and better characterized, the use of beta blockers in sepsis patients is an important research topic. This study is timely and well described. I do have some comments.

Background

- Page 7, lines 30-3: please add the reference to the recent work by Stolk et al. who report improved immune competence in septic shock patients on beta-blockers. We have included the reference although note that the paper was published after original submission of the paper to BMJOpen.
- Page 7, lines 36-40: This statement appears to reflect actual data, but the authors refer to a review. The statement is a composite of many factors that are referenced in the preceding paragraph and nicely summarised in the review. We have altered the wording of the text
- Page 8, lines 6-7: "baseline cardiovascular" should be changed to "baseline hemodynamic situation" This has been placed in the Track Changes

Methods

- Why are patients included in which noradrenaline has been stopped (<12 hours), this seems to not correspond with the inclusion criteria (i.e. requiring vasopressors)? This only applies to patients who have previously been treated with noradrenaline and then recently stopped as there is a high risk that

the noradrenaline will need to be restarted – we have made minor phrase changes to the paper to clarify this.

- The study will include patients admitted well into 2020. Will patients suffering from COVID-19 who fulfill the inclusion criteria also be included? COVID-19 has been demonstrated to have specific inflammatory properties unlike other infections/sepsis causes, which might confound results. Specific statements on this should be made in the manuscript. As the study was started in 2018 we had no plans to recruit patients for COVID 19. We have subsequently made an amendment to include COVID 19 patients but this was after the submission to BMJOpen
- Will vasopressors other than noradrenaline be used in the patients included during this trial? Yes, this is a pragmatic, real world study. The concomitant use of vasopressin is assumed. Although this does risk reducing the dose of noradrenaline and potentially reducing the number of patient eligible for the study, our hypothesis is about beta blocker immunomodulation and protection of the adverse effects of noradrenaline

Outcome measures

- Use of beta-blockers was associated with an enhanced TNF/IL-10 ratio in septic shock patients in the aforementioned recent study by Stolk et al. Therefore, I would advise to add this specific ratio to the outcome measures. Furthermore, noradrenaline inhibited production of CXCL-10 and MCP-1 in humans in vivo in this study. Therefore, I would advise to add these chemokines to the list of inflammatory markers of interest.
- It would be nice to add secondary ICU-acquired infections as an outcome measure.
- If possible, measurements of monocytic HLA-DR expression could be added (in a subset of patients).

Thank you for these excellent suggestions – we will include them in the secondary measures of our parallel mechanistic study. We are slightly constrained by our ability to make wholesale changes (for example to the Outcomes) as this study has already been recruiting since March 2018 but have done what we can to accommodate the reviewers' suggestions. The outcomes have already gone through funding, regulatory and ethical approval. We will, however, seek appropriate regulatory approval to add these to our mechanism study. We will be publishing a lab investigation protocol

Statistical analysis

- Heart rate is not part of the SOFA score and SOFA therefore represents a nicely chosen primary outcome. Thank you
- Will the detailed statistical plan be published before study inclusion is complete? Could the SAP be published as a supplement? We are publishing a separate SAP

Reviewer: 2

Reviewer Name: Lex van Loon

Institution and Country

Australian National University, Canberra, Australia

Please state any competing interests or state 'None declared':

None declared

Comments to the Author

Highly relevant study. Best of luck with the study. Looking forward to the results.

Many Thanks

Reviewer: 3

Reviewer Name

FILIPPO SANFILIPPO

Institution and Country

Azienda Ospedaliero Universitaria Policlinico San Marco, Catania, Italy

Please state any competing interests or state 'None declared':

None

Comments to the Author

Dear Editor,

Thanks for the opportunity to review the protocol of the "Study into the Reversal of Septic Shock with landiolol (Beta Blockade): Study Protocol for the STRESS-L Randomised Trial". The study is promoted by a knowledgeable panel of authors and I applaud their efforts.

Here my suggestions:

- I encourage the authors mention the results of the trial recently published in Lancet Resp Medicine 2020 <https://pubmed.ncbi.nlm.nih.gov/32243865/> Thank you – we have included this reference but please note that it was published after our paper was submitted to BMJOpen and are pleased to have the opportunity to update our reference list
- Reference 13 (review published in 2009) may be substituted by a more recent and systematic review on beta-blockade in sepsis (i.e. <https://pubmed.ncbi.nlm.nih.gov/26121122/>) We apologise for this oversight and have included this important systematic review
- Primary Objective. Please be more specific when mentioning "the efficacy". One can think of many.... Reducing SOFA, HR, other? in which timeframe? I see you later explained it, but it is better to be specific from the beginning. We are more specific in our Primary Outcome section. The Objectives of the trial remain the same as these are a composite of the things that the reviewer mention. A clinical trial is harder to run and to analyse with a composite outcome or with multiple outcomes.
- Secondary objective. Please change "improves" with "reduces". It is more appropriate for mortality and LOS. Please see above
- Regarding the "second bullet" of secondary objectives. It is better to specify if you have an hypothesis on that, explaining what exactly you suppose to find on metabolism, inflammation and cardiac samage Please see above
- Study setting. I would explain readers why you are publishing a study protocol after 2 and half years.
- Outcome measures. Please explain how will you handle (statistically) the SOFA score for deceased or discharged patients We plan to publish a SAP separately
- Please explain why you have not considered echocardiographic evaluation as part of the study objectives. The evaluation of cardiac performance is of utmost importance when considering beta-blockade, though the identification of beta-blocker responders (non-compensatory tachycardia) and non-responders (compensatory for 'fixed' stroke volume [SV]) is challenging. Echocardiography may be particularly helpful in understanding the effects of beta-blockers in septic shock patients. Even if not performing echocardiography, other measures may be very interesting. Recently Prof Morelli in a post-hoc analysis tested the ability of the difference between systolic and diastolic pressure (SDPdifference), which reflects the coupling between myocardial contractility and a given afterload, in discriminating the origin of tachycardia. The authors found that a decrease in SDPdifference could discriminate between compensatory and non-compensatory tachycardia, revealing a covert loss of myocardial contractility not detected by conventional echocardiographic parameters and deteriorating after HR reduction with esmolol. Link: <https://pubmed.ncbi.nlm.nih.gov/32690246/> Performing ECHO is easy in a single centre setting but validating training and unifying ECHO techniques and interpretation across 40 centres whilst minimising inter- and intra- user variability would involve

training and validation that was not funded. We do have smaller sub-studies that are running outside of the main study presented here

VERSION 2 – REVIEW

REVIEWER	Peter Pickkers Radboudumc, The Netherlands
REVIEW RETURNED	14-Jan-2021

GENERAL COMMENTS	The authors have thoughtfully and thoroughly revised the manuscript which we recommend is acceptable for publication. We have one minor point to make: apparently COVID-19 patients who fulfill the inclusion criteria are included in the trial. However, the manuscript does not mention whether these patients are analysed as a separate subgroup or indiscriminately from other sepsis patients. As it is a vastly different disease from an inflammatory standpoint we would suggest to include as many sepsis patients with other etiologies as possible, in order to expand the external validity beyond 2020-2021.
---

REVIEWER	FILIPPO SANFILIPPO Azienda Ospedaliera Policlinico San Marco, Catania (Italy)
REVIEW RETURNED	18-Dec-2020

GENERAL COMMENTS	Thank you for addressing my concerns and for explaining the delay in publishing this protocol. I wish you can finish the study as soon as feasible, providing interesting results to the entire scientific community.
---